# The effects of vaginal gel from Myrtus communis on the sexual function of married women during reproductive aging: A study protocol for a randomized controlled trial

Bahareh khajehpour[1], Raheleh Babazadeh[2,3*], Roshanak Salari[4], Jamshid Jamali[5], Malihe Mahmoudinia[6]

1 MSc Student of Midwifery, Faculty of Nursing and Midwifery, Mashhad University of Medical Sciences, Mashhad, Iran, 2 Nursing and Midwifery Care Research Center, Mashhad University of Medical Sciences, Mashhad, Iran, 3 Nursing and Midwifery Care Research Center, Mashhad University of Medical Sciences, Mashhad, Iran, 4 Associate Professor of Drug Control, Department of Pharmaceutical Sciences in Persian Medicine, School of Persian and Complementary Medicine, Mashhad University of Medical Sciences, Mashhad, Iran, 5 Associate Professor of Biostatistics, Department of Biostatistics, School of Health, Mashhad University of Medical Sciences, Mashhad, Iran, 6 Department of Obstetrics and Gynecology, Faculty of Medicine, Maryland, Associate Professor of Obstetrics & Gynecology, Fellowship of infertility, Mashhad University of Medical Sciences, Mashhad, Iran

* 114ram114@gmail.com, babazadehr@mums.ac.ir

## Abstract

### Introduction

Human sexuality is an important aspect of functionality, and many patients believe that it determines their quality of life. Sexual dysfunction can lead to stress, strained relationships, and a decrease in self-esteem. The majority of modern medical treatments for improving female sexual function are associated with significant side effects and high costs. Traditionally, M. communis has been used to treat sexual impotence. Here, we present the protocol of an interventional clinical phase II study to test the hypothesis that vaginal gel containing Myrtus communis extract can improve sexual function in women of reproductive age.

### Methods/Design

In a prospective, randomized, placebo-controlled and highly blind clinical phase II trial, 80 women aged 18--40 years with sexual dysfunction meeting the inclusion criteria will be randomized to an intervention group receiving a vaginal gel containing myrtle leaf extract (n = 38) or an active control group receiving a placebo gel with an identical appearance (n = 38). Randomization will be performed via a permuted block technique with random allocation software. This study will be conducted at the women's clinic of Imam Reza Hospital in Mashhad. Blinded assessments of outcome

**Data availability statement:** Deidentified research data will be made publicly available when the study is completed and published.

**Funding:** The EVGMCSFW trial is funded by a grant from Mashhad University of Medical Sciences.

**Competing interests:** The authors have declared that no competing interests exist.

**Abbreviations:** FSDs, Female sexual dysfunction; HSDD, Hypoactive sexual desire disorder; FOD, Female orgasmic disorder; EVGMCSFW, Effects of vaginal gel from Myrtus communis on the sexual function of women; NVD, Natural vaginal delivery.

variables will be conducted twice: before treatment and one month after treatment. The primary outcome measure was sexual function.

## Discussion

This randomized controlled clinical trial evaluates the efficacy of the vaginal gel Myrtus communis on the sexual function of married women during reproductive age. The study design presented here fulfills the criteria of a high-quality clinical phase II trial of sexual function.

## Trial registration

IRCT.behdasht.gov.ir Identifier: IRCT20230723058892N1

---

## Introduction

The term "sexual function" describes how the body reacts at different points in the sexual response cycle [1]. Human sexuality is an important aspect of functionality, and many patients believe that it determines their quality of life [2]. A vital component of health and wellbeing is sexuality. The World Health Organization defines safe sexual experiences as a state of physical, emotional, mental, and social well-being related to sexuality, not just the absence of sexual dysfunction [3]. Complex interactions between biological and psychosocial factors determine the human sexual response. Depending on the time, place, and situation, these variables can differ among cultures, people, and even within the same person. To help with the conceptualization, diagnosis, and treatment of sexual disorders in both men and women, various models have been proposed that attempt to characterize these complexities [4]. According to recent proposals, sexual function can be viewed as a circuit consisting of four primary domains: libido, arousal, orgasm, and satisfaction. Each element may influence others in a positive or negative way, or they may overlap [5]. An important sign of a successful marriage, which in turn ensures the survival and well-being of the family, is sexual satisfaction, or, to put it another way, the satisfaction of that person's sex claim. It can be described as experiencing gratifying and enjoyable sexual experiences, such as an exhilarating orgasm and a good and positive feeling [6]. It was defined as "an effective response arising from one's subjective evaluation of the positive and negative dimensions associated with one's sexual relationship" by Lawrance and Byers (1995), who offered one of the most widely accepted definitions. It also has a significant effect on people's general quality of life [7]. Higher levels of sexual satisfaction are linked to more stable relationships and improved mental and physical health [8]. The hallmarks of sexual dysfunctions include ongoing and frequent challenges in accessing and completing one or more stages of the physical sexual response (i.e., resolution, orgasm, arousal, and desire) [9]. There are limited data on the incidence and prevalence of FSDs. The definitions of sexual dysfunction, the various diagnostic categories used, the makeup of the sample populations, and methods of data collection all cause significant differences

in the available data [4]. Between 30% and 50% of American women suffer from female sexual dysfunction, which is age-related, progressive, and extremely common [5]. According to reports, the prevalence varies from 25-91% in different countries [10]. Studies in Iran have shown that approximately one-third of women of reproductive age experience sexual disorders [11]. Additionally, the overall prevalence of sexual dysfunction among women has been reported to be 43.9% [12]. In the PRESIDE study (Prevalence of Female Sexual Problems Associated with Distress and Determinants of Treatment Seeking), which included 50,001 US women with a 63% response rate, the prevalence of HSDD was 8.9% in women between the ages of 18 and 44, 12.3% in women between the ages of 45 and 64, and 7.4% in women 65 and older. Sixty-four percent of women had desire difficulties, 31 percent had arousal difficulties, 35 percent had orgasm difficulties, and 26 percent experienced sexual pain, according to a 2006 review by Hayes and colleagues [4]. FOD and other forms of sexual dysfunction can result from a variety of biopsychosocial factors. Conditions such as diabetes, neurological disorders, cardiovascular disease, unhealthy lifestyles, the effects of biological factors, and alcohol and drug abuse are examples of pharmacological agents. While sociological factors may include things such as unemployment, financial difficulties, and civil status (being single), psychological factors include things such as stress, depression, personal problems, body image issues, and relationship problems [13]. Whatever the cause, there are many detrimental effects of sexual dysfunction and the ensuing decline in sexual satisfaction. These include increased social problems such as crime, assault, infidelity, divorce, and psychological disorders and can negatively impact an individual's overall functioning [14]. There is evidence of a strong correlation between emotional and physical disorders and sexual dysfunction. Sexual dysfunction can lead to stress, strained relationships, and a decrease in self-esteem. Sexual problems can cause serious problems and even the dissolution of a marriage, and sexual problems are a crucial aspect of marital satisfaction [15]. Dissatisfaction with sexual relationships has been found to be the cause of 50–60% of divorces and 40% of infidelity and secret relationships in couples [16]. The evaluation should include a complete medical, psychosocial, and sexual history and physical examination, including a gynecologic examination. Therapy is mostly determined by etiology, but it frequently includes behavior modification, psychotherapy, education, and sometimes pharmacotherapy. It is best to take a multidisciplinary approach [17]. Owing to their lower costs and fewer side effects, many herbal remedies have been investigated for their potential to improve female sexual function. In contrast, the majority of modern medical treatments are associated with significant side effects and high costs [18]. Plants that have been used to treat sexual disorders include pomegranates, ginger, ginseng, black cumin, and some pistachio species [19]. In this context, some studies have demonstrated the effectiveness of these plants [18]. Myrtle (Myrtus communis L., Myrtaceae) is a well-known medicinal plant that has been used worldwide in traditional medicine. Evergreen leaves, such as those of myrrh or eucalyptus, are aromatic when crushed and range in length from 2--5 cm. Its astringency is primarily responsible for its intense and bitter flavor [20]. Traditional medicine uses it to treat patients because of its therapeutic and aromatic properties. The essential oils and compounds present in its leaves and fruit are responsible for its medicinal applications [21]. Its astringent, tonic, and antiseptic properties support its application in the treatment of wounds and digestive and urinary system disorders. Traditionally, M. communis has been used to treat sexual impotence [20]. The increase in testosterone caused by Myrtus communis leaf extract is likely due to the presence of compounds such as flavonoids (which inhibit 5 alpha reductase and aromatase), ascorbic acid, myricetin (which inhibits aromatase), and linoleic, oleic, and palmitic acids, which are compressor 5 alpha reductase, 1,8 cineol, and delta cadinene (Cytochrome 480) [22]. Increasing testosterone production also enhances the metabolism of steroids [23]. Additionally, the leaves contain tannins; flavonoids such as quercetin and catechin; myricetin derivatives; and coumarins [20]. This plant's flavonoids, especially quercetin, have phytoestrogenic effects. These phytoestrogens improve lubrication, sexual arousal, and sexual desire, all of which contribute to more fulfilling orgasm [24]. Additionally, tannins are bitter, astringent plant polyphenols that can shrink or bind proteins and precipitate them. Consequently, tannins tighten and contract muscles, which in turn causes the vagina to tighten as well, potentially improving the orgasm [25]. Given the high prevalence of sexual dysfunction among Iranian women and considering the mentioned benefits of the Myrtus communis extract, its affordability, availability, and lack of potentially harmful

side effects when it is prescribed at appropriate doses and the absence of studies examining the effectiveness of this plant on women's sexual function, this study was designed to assess the effect of Myrtle vaginal gel on the sexual function of married women of reproductive age in Mashhad.

## Methods/Design

### Study design

The EVGMCSFW is a prospective, parallel group, randomized, placebo-controlled, superiority and highly blind phase II clinical trial for the evaluation of the efficacy of a vaginal gel of Myrtus communis on the sexual function of married women of reproductive age compared with an active control group.

### Study setting

This study was conducted at the Obstetrics and Gynecology Clinic at Emam Reza Hospital.

### Recruitment

Recruitment for this study began in April 2024 and was expected to be completed by April 2025. Data collection was anticipated to conclude by May 2025. The results were expected to be available by June 2025. The study was approved by the Ethics Committee of Mashhad University of Medical Sciences on 22 July 2023. Patients were recruited consecutively during regular consulting hours. Recruitment strategies involve active recruitment by information flyers and posters or by informing patients about the study.

### Drug formulation

The vaginal gel and placebo were standardized, prepared, and packaged at the Department of Pharmaceutical Sciences in Persian Medicine, School of Persian and Complementary Medicine, Mashhad University of Medical Sciences. A myrtle sample was collected from Mashhad, Khorasan Razavi Province, Iran. A herbarium sample was prepared and sent to the Herbarium Center of Mashhad University of Medical Sciences. To prepare the aqueous extract, dried myrtle powder was obtained. The powdered myrtle was immersed in an aqueous solution and placed in a water bath for half a day. The filtered aqueous extract was placed in an incubator for two days. Ultimately, 290 grams of extract were obtained from 3500 grams of myrtle leaves. Subsequently, a vaginal gel was formulated based on Carbomer at a 2% concentration. Finally, the extract, at a 5% concentration, was incorporated into the carbomer base. According to the study by Mahboubi et al., the oral toxic dose is approximately 56 grams, which is substantially higher than the administered dosage in this formulation, confirming its safety [26].

### Standardization of the extracts of Myrtus communis

The extracts of Myrtus communis was standardized based on content of phenolic compounds. A 20 µL sample of the plant extract (10 mg/mL) or gallic acid as the standard (50–500 mg/L) was combined with 100 µL of Folin-Ciocalteu reagent and 300 µL of a sodium carbonate solution (1 mol/L). The mixture volume was adjusted to 2 mL using deionized water. After 2 hours, the absorbance was recorded at 765 nm using a spectrophotometer. A standard curve for gallic acid was generated, and the phenolic content of the extracts was quantified as milligrams of gallic acid equivalents [27] (Fig 1).

### Content of phenolic compounds in Myrtus communis

The content of total phenols in the extract of Myrtus communis was 196 mg gallic acid equivalent per gram of the extract.

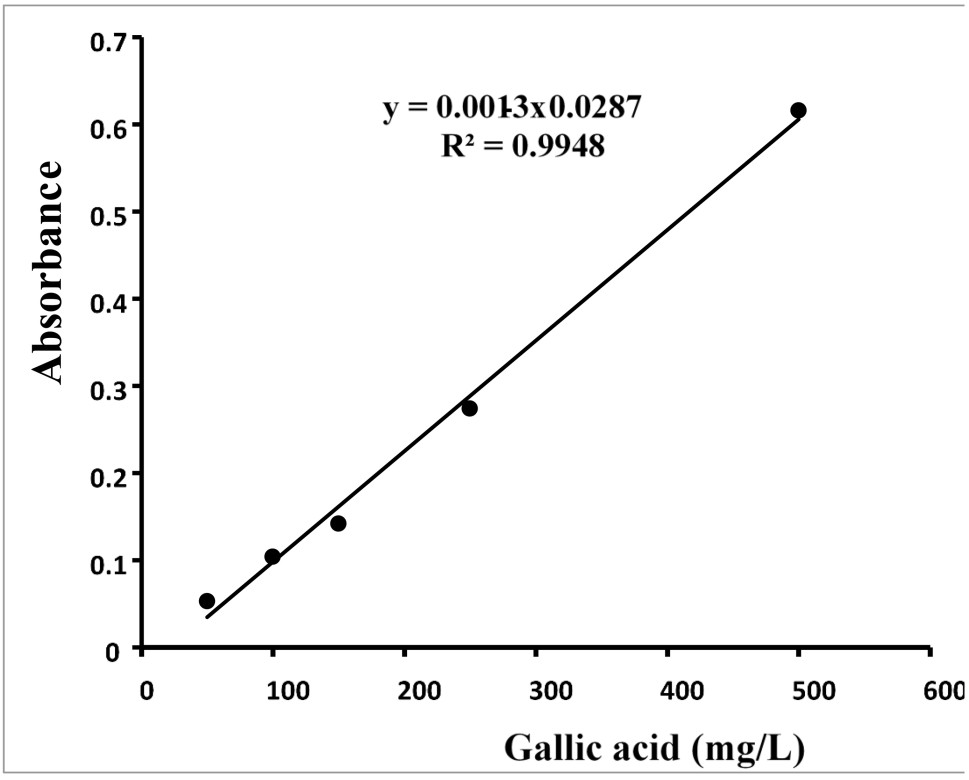

**Standard curve**

$y = 0.0013x\,0.0287$

$R^2 = 0.9948$

Absorbance of the extract = 2.520

**Fig 1. Standare curve.**

## Participants

The inclusion criteria for participation in the EVGMCSFW trial include being a married woman, being aged 18--40 years, providing informed consent to participate in research, having a score of less than 28 on the Female Sexual Function Index (FSFI) questionnaire [28,29], having a depression score of less than 21, having an anxiety score of less than 15, and having a stress score of less than 26 on the Depression Anxiety Stress Scale (DASS) questionnaire [30]. She had a history of NVD, a regular menstrual cycle (26–32 days) and no abnormal uterine bleeding (AUB), engaged in sexual activity at least once a week and have basic literacy skills, defined as the ability to read and understand the consent form and study questionnaires without assistance, confirmed during the enrollment process. Additionally, participants must not be pregnant or breastfeeding, and must have no history of infertility or conditions associated with vaginal atrophy (defined as no history of oophorectomy, premature menopause, pelvic radiotherapy, chemotherapy, or hormonal therapy for cancer treatment; no use of oral contraceptive pills or anti-estrogenic medications for endometriosis; and no uncontrolled diabetes). No hormone therapy was given for two months prior to the study. Additionally, the absence of any specific mental disorders in both partners and no underlying medical conditions affect sexual function in the woman or her partner.

The exclusion criteria during the study included sensitivity to plant-based substances or herbal medicines, lack of willingness to continue participation in the study, completion of sexual health education during the intervention period,

diagnosis of vaginal infections requiring treatment during the study, pregnancy during the study and experiencing an adverse or stressful event for the individual or their partner during the study.

## Sampling and randomization

The participants was selected in an easy and accessible manner. Participants meeting eligibility criteria were enrolled consecutively. Randomization was performed using a **permuted block design** with a block size of 4, generated by computer-based random allocation software (e.g., Random Allocation Software). This method ensures balanced group sizes (1:1 allocation) throughout the recruitment period. An independent statistician was generate the allocation sequence. Group assignment (Myrtus communis gel or placebo) was concealed using sequentially numbered, opaque, sealed envelopes (SNOSE) until after enrollment. Participants, care providers, outcome assessors, and data analysts was blinded to group assignment. In this study, there are six blocks: AABB (1), ABAB (2), ABBA (3), BBAA (4), BABA (5), and BAAB (6). One of the blocks was randomly selected. If the first block, AABB, was chosen, the first and second participants were assigned to Group A, and the third and fourth participants were assigned to Group B. This process continues until all participants were allocated. The key feature of this method was that both study groups have an equal number of participants. After obtaining written informed consent, the study coordinator was opened the next envelope in sequence to assign the participant to either the intervention or placebo group.

## Intervention and controls

The researcher, after obtaining approval from the Ethics Committee of Mashhad University of Medical Sciences and providing a written referral letter from the faculty, visited the research centers for data collection and sampling. After explaining the research objectives and obtaining written informed consent, each potential participant was screened against the inclusion criteria. Those who met the criteria were selected to participate in the study. Each research participant subsequently was completed the Female Sexual Function Index (FSFI) questionnaire, the Depression, Anxiety, and Stress Scale (DASS-21) questionnaire and the Most Bothersome Symptom Scale for Vaginal Atrophy (MBS) at the same location. If additional clarification was needed, the researcher provides the necessary guidance. Given that sexual issues were among the most private aspects of women's lives, the researcher established communication with the study participants in a completely private setting, ensuring the confidentiality of all the information. Each participant in the two groups was received either a 5% Myrtus communis vaginal gel or a placebo gel, both of which were identical in appearance. They were instructed to use one applicator of the assigned gel 15 minutes prior to sexual intercourse for a duration of four weeks. As shown in Fig 1, to ensure medication adherence by each research participant, evaluate their satisfaction with the received treatment, and monitor potential side effects of the medication, four weekly follow-up phone calls were conducted with each participant at the end of every week. If any adverse drug reactions or complications were arised, the participant was closely monitored and managed by the researcher and the supervising gynecologist. At the end of week 4, women were contacted via telephone and asked to visit the clinic for a follow-up evaluation. During this visit, treatment satisfaction and medication side effects were reassessed, and the FSFI (Female Sexual Function Index) questionnaire was completed.

## Outcome parameters

The primary outcome was the course of sexual function across six domains, including libido, sexual arousal, vaginal lubrication, orgasm, dyspareunia, and sexual satisfaction, comparing pre- and post-intervention (4 weeks) between the placebo and intervention groups.

   **Sample size** sample size was calculated using an effect size based on the formula, with a significance level of 0.05, 80% statistical power, and a medium effect size (d = 0.7) derived from Abbasi Pirouz et al. [31] yielding 32 participants per

group. Accounting for the interventional nature of the study and an anticipated attrition rate of approximately 20%, the final sample size was increased to 38 participants per group

$$n = \frac{2\left(z_{1-\frac{\alpha}{2}} + z_{1-\beta}\right)^2}{f^2} = \frac{2(1.96 \cdot 0.84)^2}{0.7^2} = 32$$

## Blinding

Neither the patients, the researchers, nor the statistical consultants were aware of the treatment assignment in this study. This triple-blind design was helped minimize bias and ensure the objectivity of the results. The gel and placebo were made identical in appearance and randomly assigned to groups 'A' and 'B' by a third party to maintain blinding. Treatment blinding was remained concealed until the end of data collection and statistical analysis. An easy unblinding procedure was allowed for rapid unblinding of a patient in case of medical necessity. Unblinding inevitably results in the exclusion of the respective patient.

**Adherence Monitoring & Missing Data Management** Adherence to the gel regimen was assessed through:

1. Daily Diaries: Participants were recorded gel application.

2. Returned Gel Weight: Used gel tubes were collected and weighed to quantify usage.

3. Self-Report: Participants were queried about adherence during follow-up.

   Adherence wasdefineded as using ≥80% of the prescribed doses. Adherence rates were reported per group.

## Ethical Considerations

All participants were provided with verbal and written information regarding the study, including its purpose, procedures, potential risks and benefits, voluntary participation, and the right to withdraw at any time. Written informed consent was obtained before data collection. The consent form was written in simple language and includes contact details of the research team. The study protocol had been approved by the Ethics Committee of Mashhad University of Medical Sciences, and the confidentiality of participants were maintained throughout the study. As one of the inclusion criteria was basic literacy, individuals with limited reading and writing abilities were not enrolled in the study. Consequently, challenges related to obtaining informed consent from participants with limited literacy did not arise.

All participant data was handled confidentially. Only the research team had accessed to the data, which was stored in password-protected files. All reports and publications resulting from this study will use aggregated data and will not contain any information that could identify individual participants. If any participant was reported distressing or unexpected symptoms related to sexual function or mental health during the study, she was referred to an appropriate healthcare provider for further evaluation and care. The study team were monitored participant safety throughout the study period and take necessary action if any adverse events was occured.

## Statistical methods

After data collection and entry into SPSS, the data were analyzed descriptively and inferentially. Descriptive statistics were presented via tables, graphs, and appropriate summary measures. For continuous variables, independent samples t-tests/Mann–Whitney U tests were used to compare groups, and paired t-tests/Wilcoxon signed–rank tests were used for paired comparisons. For categorical variables, the chi-square test was used. The normality of continuous variables were assessed via the Kolmogorov–Smirnov test. Other statistical tests, such as repeated-measures ANOVA and regression analysis, may be used as needed. The level of significance was set at 0.05.

Primary Analysis: Intention-to-treat (ITT) analysis was performed, including all randomized participants in their originally assigned groups, regardless of adherence or dropout.

Missing Data Handling (ITT): For the primary outcome (FSFI change), missing postintervention data were addressed using multiple imputation by chained equations (MICE), creating 20 imputed datasets. The imputation model was included baseline FSFI score, age, group assignment, and relevant covariates showing association with amusingness. Sensitivity analyses were compared results to complete-case analysis.

Supportive Analysis: A per-protocol (PP) analysis was also conducted, including only participants meeting predefined adherence criteria (≥80% adherence) and completing the 1-month assessment. Results from ITT and PP analyses were compared to assess robustness.

## Discussion

Female sexual dysfunction (FSD) has often been overlooked and inadequately addressed, primarily due to the absence of clear definitions, reliable evaluation tools, and effective treatment options. FSD is a significant public health issue that profoundly affects patients' quality of life [32]. Studies in Iran have shown that approximately one-third of women of reproductive age experience sexual disorders [11]. Given the high prevalence of sexual dysfunction among Iranian women and considering the reported properties of myrtle leaf extract, its affordability, accessibility, and lack of significant adverse effects when it is administered at appropriate doses, the proposed study has the potential to significantly contribute to the evaluation of the efficacy of this plant in patients with sexual dysfunction. The primary outcome measure was sexual function. A triple-blind randomized controlled study design was selected to minimize bias while enhancing the accuracy of the results. The study design presented here fulfills the criteria of a high-quality clinical phase

II trial of sexual function. If the study results are positive, this method could be introduced as a complementary standard approach for improving sexual dysfunction and reducing its associated complications in patients suffering from this condition.

### Trial status

The EVGMCSFW trial is currently in the data analysis phase.

### Supporting information

**S1 Fig. Timeline of study.**
(DOCX)

### Acknowledgments

We are grateful to the Dr. Alireza Ataei and Mr. Yasser Jeddi regarding assistance in completing various stages of study.

### Author contributions

**Conceptualization:** Bahareh Khajehpour, malihe mahmoudinia, Roshanak salari, raheleh babazadeh.

**Data curation:** Bahareh Khajehpour, malihe mahmoudinia.

**Formal analysis:** jamshid jamali.

**Methodology:** Roshanak salari, jamshid jamali, raheleh babazadeh.

**Supervision:** raheleh babazadeh.

**Writing – original draft:** Bahareh Khajehpour.

**Writing – review & editing:** Roshanak salari, jamshid jamali, raheleh babazadeh.

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
