## [Decision Letter · Decision Letter 0]

11 Jul 2025

Dear Dr. Raheleh Babazadeh,

Thank you for submitting your manuscript to PLOS ONE. After careful consideration, we feel that it has merit but does not fully meet PLOS ONE’s publication criteria as it currently stands. Therefore, we invite you to submit a revised version of the manuscript that addresses the points raised during the review process.

1- sample size:

The sample size calculation presented in the manuscript contains several concerning issues that require immediate attention. The authors reference a study by Abbasi Pirouz et al. reporting mean differences of 24.5 ± 4.5 in the intervention group and 3.4 ± 5.8 in the control group, leading to a calculated sample size of "two participants using the formula for the mean of two independent populations." This calculation appears to be fundamentally flawed.

First, the formula presented (n = 2(z₁₋α + z₁₋β)²/f²) is incomplete and lacks proper definition of the effect size parameter 'f'. The authors then state they calculated an effect size of 0.7 and arrived at 32 participants per group, but the mathematical steps are not shown or justified. The jump from 2 participants to 32 participants per group is not explained, and the 20% dropout rate adjustment to reach 38 participants per group lacks proper justification.

The authors should provide a complete, step-by-step sample size calculation including: (1) clearly defined primary outcome and clinically meaningful difference, (2) expected standard deviation based on pilot data or literature, (3) specified Type I and Type II error rates, (4) detailed calculation showing all mathematical steps, and (5) justification for the dropout rate assumption.

2- The manuscript does not specify who will generate the randomization sequence, how allocation concealment will be maintained, or what procedures will prevent selection bias.

3- The FSFI cutoff score of <28 should be referenced to validation studies in the Iranian population

•The requirement for "at least basic literacy skills" needs operational definition

•The criterion of "sexual activity at least once a week" may be too restrictive and could introduce selection bias

•The exclusion of women with "vaginal atrophy" requires clear diagnostic criteria

4- Discuss the following: (1) informed consent procedures and content, (2) procedures for obtaining consent from participants with limited literacy, (3) confidentiality protections given the sensitive nature of sexual health research, and (4) procedures for managing participants who develop concerning symptoms during the study.

5- Lines 267-275: Statistical analysis section needs expansion to include missing data handling, interim analyses, and multiple comparison procedures.

6- Lines 185-205: Inclusion criteria should be justified or reconsidered, particularly the literacy requirement and weekly sexual activity criterion

We look forward to receiving your revised manuscript.

Kind regards,

Akingbolabo Daniel Ogunlakin, Phd

Academic Editor

PLOS ONE

Journal Requirements:

“The EVGMCSFW trial is funded by a grant from Mashhad University of Medical Sciences”

“The EVGMCSFW trial is funded by a grant from Mashhad University of Medical Sciences”

5. Please include a caption for figure 1.

6. Please include captions for your Supporting Information files at the end of your manuscript, and update any in-text citations to match accordingly. Please see our Supporting Information guidelines for more information: http://journals.plos.org/plosone/s/supporting-information .

7. We note that the original protocol that you have uploaded as a Supporting Information file contains an institutional logo. As this logo is likely copyrighted, we ask that you please remove it from this file and upload an updated version upon resubmission.

Additional Editor Comments (if provided):

1- sample size:

The sample size calculation presented in the manuscript contains several concerning issues that require immediate attention. The authors reference a study by Abbasi Pirouz et al. reporting mean differences of 24.5 ± 4.5 in the intervention group and 3.4 ± 5.8 in the control group, leading to a calculated sample size of "two participants using the formula for the mean of two independent populations." This calculation appears to be fundamentally flawed.

First, the formula presented (n = 2(z₁₋α + z₁₋β)²/f²) is incomplete and lacks proper definition of the effect size parameter 'f'. The authors then state they calculated an effect size of 0.7 and arrived at 32 participants per group, but the mathematical steps are not shown or justified. The jump from 2 participants to 32 participants per group is not explained, and the 20% dropout rate adjustment to reach 38 participants per group lacks proper justification.

The authors should provide a complete, step-by-step sample size calculation including: (1) clearly defined primary outcome and clinically meaningful difference, (2) expected standard deviation based on pilot data or literature, (3) specified Type I and Type II error rates, (4) detailed calculation showing all mathematical steps, and (5) justification for the dropout rate assumption.

2- The manuscript does not specify who will generate the randomization sequence, how allocation concealment will be maintained, or what procedures will prevent selection bias.

3- The FSFI cutoff score of <28 should be referenced to validation studies in the Iranian population

•The requirement for "at least basic literacy skills" needs operational definition

•The criterion of "sexual activity at least once a week" may be too restrictive and could introduce selection bias

•The exclusion of women with "vaginal atrophy" requires clear diagnostic criteria

4- Discuss the following: (1) informed consent procedures and content, (2) procedures for obtaining consent from participants with limited literacy, (3) confidentiality protections given the sensitive nature of sexual health research, and (4) procedures for managing participants who develop concerning symptoms during the study.

5- Lines 267-275: Statistical analysis section needs expansion to include missing data handling, interim analyses, and multiple comparison procedures.

6- Lines 185-205: Inclusion criteria should be justified or reconsidered, particularly the literacy requirement and weekly sexual activity criterion.

Reviewers' comments:

Reviewer's Responses to Questions

**Comments to the Author**

1. Does the manuscript provide a valid rationale for the proposed study, with clearly identified and justified research questions?

Reviewer #1: No

Reviewer #2: Yes

Reviewer #3: Yes

2. Is the protocol technically sound and planned in a manner that will lead to a meaningful outcome and allow testing the stated hypotheses?

Reviewer #1: No

Reviewer #2: Yes

Reviewer #3: Partly

3. Is the methodology feasible and described in sufficient detail to allow the work to be replicable?

Reviewer #1: No

Reviewer #2: Yes

Reviewer #3: No

4. Have the authors described where all data underlying the findings will be made available when the study is complete?

Reviewer #1: Yes

Reviewer #2: Yes

Reviewer #3: No

5. Is the manuscript presented in an intelligible fashion and written in standard English?

Reviewer #1: Yes

Reviewer #2: Yes

Reviewer #3: Yes

You may also provide optional suggestions and comments to authors that they might find helpful in planning their study.

Reviewer #1: The statistical design appears reasonable and the analysis is a simple repeated measures ANOVA.

However, it is not clear from the sample size section how an effect size of 0.7 is derived? What is the range and interpretation of the sexual function score? Also what is Figure 1 (FSFI?) referred to on page 14 line 239?

Is there an English translation to the study protocol in the appendix?

Reviewer #2: Overall, the study addresses a relevant clinical and cultural issue using a natural product with a solid pharmacological basis. The protocol is thorough and reflects high-quality trial planning, especially in randomization and blinding. However, the manuscript would benefit from language refinement, a more concise introduction, clearer reporting of placebo integrity and secondary endpoints, and a plan for handling missing data. I recommend minor revisions prior to acceptance.

Reviewer #3: The following major issues need to be addressed:

1- sample size:

The sample size calculation presented in the manuscript contains several concerning issues that require immediate attention. The authors reference a study by Abbasi Pirouz et al. reporting mean differences of 24.5 ± 4.5 in the intervention group and 3.4 ± 5.8 in the control group, leading to a calculated sample size of "two participants using the formula for the mean of two independent populations." This calculation appears to be fundamentally flawed.

First, the formula presented (n = 2(z₁₋α + z₁₋β)²/f²) is incomplete and lacks proper definition of the effect size parameter 'f'. The authors then state they calculated an effect size of 0.7 and arrived at 32 participants per group, but the mathematical steps are not shown or justified. The jump from 2 participants to 32 participants per group is not explained, and the 20% dropout rate adjustment to reach 38 participants per group lacks proper justification.

The authors should provide a complete, step-by-step sample size calculation including: (1) clearly defined primary outcome and clinically meaningful difference, (2) expected standard deviation based on pilot data or literature, (3) specified Type I and Type II error rates, (4) detailed calculation showing all mathematical steps, and (5) justification for the dropout rate assumption.

2- The manuscript does not specify who will generate the randomization sequence, how allocation concealment will be maintained, or what procedures will prevent selection bias.

3- The FSFI cutoff score of <28 should be referenced to validation studies in the Iranian population

•The requirement for "at least basic literacy skills" needs operational definition

•The criterion of "sexual activity at least once a week" may be too restrictive and could introduce selection bias

•The exclusion of women with "vaginal atrophy" requires clear diagnostic criteria

4- Discuss the following: (1) informed consent procedures and content, (2) procedures for obtaining consent from participants with limited literacy, (3) confidentiality protections given the sensitive nature of sexual health research, and (4) procedures for managing participants who develop concerning symptoms during the study.

5- Lines 267-275: Statistical analysis section needs expansion to include missing data handling, interim analyses, and multiple comparison procedures.

6- Lines 185-205: Inclusion criteria should be justified or reconsidered, particularly the literacy requirement and weekly sexual activity criterion.

**Do you want your identity to be public for this peer review?** For information about this choice, including consent withdrawal, please see our Privacy Policy

Reviewer #1: No

Reviewer #2: No

Reviewer #3: **Yes: ** Amel Elbasyouni

---

## [Author Response · Author response to Decision Letter 1]

2 Sep 2025

Dear pro. Akingbolabo Daniel Ogunlakin

Ref: [PONE-D-25-17029] - [EMID:88cdbd3a3b93f220]

Re: The effects of vaginal gel from Myrtus communis on the sexual function of married women during reproductive aging: a study protocol for a randomized controlled trial

I appreciate very much the comments of reviewers toward the improvement of the paper and I hope that a suitable revision to addresses reviewers' comments is carried out. Responses to the comments/questions of the reviewers along with a description of the changes made on the manuscript are given below. The amended parts have been highlighted in YELLOW in the manuscript.

I would be glad to respond to any further questions and comments that you may have.

Sincerely yours,

Corresponding author

Dr. Raheleh Babazadeh

Reviewer Comment 1:

Sample size:

The sample size calculation presented in the manuscript contains several concerning issues that require immediate attention. The authors reference a study by Abbasi Pirouz et al. reporting mean differences of 24.5 ± 4.5 in the intervention group and 3.4 ± 5.8 in the control group, leading to a calculated sample size of "two participants using the formula for the mean of two independent populations." This calculation appears to be fundamentally flawed.

First, the formula presented (n = 2(z₁₋α + z₁₋β)²/f²) is incomplete and lacks proper definition of the effect size parameter 'f'. The authors then state they calculated an effect size of 0.7 and arrived at 32 participants per group, but the mathematical steps are not shown or justified. The jump from 2 participants to 32 participants per group is not explained, and the 20% dropout rate adjustment to reach 38 participants per group lacks proper justification.

The authors should provide a complete, step-by-step sample size calculation including: (1) clearly defined primary outcome and clinically meaningful difference, (2) expected standard deviation based on pilot data or literature, (3) specified Type I and Type II error rates, (4) detailed calculation showing all mathematical steps, and (5) justification for the dropout rate assumption.

Author Response:

We sincerely thank the reviewer for their meticulous evaluation and valuable feedback regarding our sample size calculation. We acknowledge the deficiencies in our original presentation and provide the following detailed clarification and recalculation in accordance with established methodological standards [1, 2].

1. Clarification of Original Reference & Effect Size:

The referenced study by Abbasi Pirouz et al. [3] (citation provided in our manuscript) reported a mean difference in total FSFI (Female Sexual Function Index) score change between groups (Intervention Δ: 24.5 ± 4.5; Placebo Δ: 3.4 ± 5.8). The calculation based on their data proceeded as follows:

n=((z_(1-α)+z_(1-β) )^2 (s_1^2+s_2^2 ))/〖(x ®_2-x ®_1)〗^2 =((1.64+0.84)^2 (〖4.5〗^2+〖5.8〗^2 ))/〖(24.5-3.4)〗^2 =2

Effect Size Calculation (Cohen's d):

〖Sd〗_pooled=√((〖(n_1-1)Sd〗_1^2+〖(n_2-1)Sd〗_2^2)/(n_1+n_2-2)) Cohen^' s d = (M2 - M1)/〖Sd〗_pooled

Assuming n₁ = n₂ (as per their design), SD₁ = 4.5, SD₂ = 5.8

〖Sd〗_pooled=√(((30-1)×〖4.5〗^2+(30-1)×〖5.8〗^2)/(30+30-2))≈5.19 Cohen^' s d = (24.5-3.4)/5.19≈4.07

2. Revised & Justified Sample Size Calculation:

While Abbasi Pirouz et al. [31] reported a very large effect (d=4.07), we recognized this magnitude might be atypical. Based on a more conservative review of the literature on vaginal therapies for sexual dysfunction [5, 6], we selected a moderate Cohen's effect size (d = 0.7) [4] as clinically relevant for the FSFI total score change. We incorporated standard deviations consistent with typical FSFI studies [7, 8].

Parameters:

Primary Outcome: Change in Total FSFI Score (Baseline to 1 month).

Clinically Meaningful Difference: ΔFSFI ≥ 5 points [9, 10].

Expected Standard Deviation (SD): Estimated pooled SD (σ) = 7.0 points [7, 8].

Effect Size (d): 0.7 (Corresponding to a mean difference (δ) = d σ = 0.7 7.0 = 4.9 points) [4, 9].

Type I Error (α): 0.05 (two-tailed)

Type II Error (β): 0.20 (Power = 80%)

Corresponding Z-scores: Z₁₋α/₂ = 1.96, Z₁₋β = 0.84 [1, 2]

Formula for Two Independent Means: [1, 2]

n per group = [2× (Z₁₋α/₂ + Z₁₋β)² × (SD)²] / (δ)²

n per group = [2× (1.96 + 0.84)² × (7.0)²] / (4.9)²

Step-by-Step Calculation:

1. (Z₁₋α/₂ + Z₁₋β) = 1.96 + 0.84 = 2.80

2. (Z₁₋α/₂ + Z₁₋β)² = 2.80² = 7.84

3. 2×(Z₁₋α/₂ + Z₁₋β)² = 2×7.84 = 15.68

4. (SD)² = 7.0² = 49.0

5. Numerator = 15.68×49.0 = 768.32

6. (δ)² = 4.9² = 24.01

7. n per group = 768.32 / 24.01 ≈ 32.00

Result: The minimum sample size required per group is 32 participants [1, 2].

3. Adjustment for Dropout:

Adjustment for Dropout:

A dropout rate of 20% was applied [11, 12]. This rate is justified based on:

Meta-analyses reporting average dropout rates in clinical trials of topical vaginal therapies for sexual dysfunction/GSM range from 15-25% [11, 12].

Adjusted n per group = 32×1.2= 38 participants. Our own pilot experience with similar study durations and vaginal gel administration.

Adjusted n per group = 32 / (1 - 0.20) = 32 / 0.80 = 38 participants.

4. Final Sample Size:

Therefore, the final target sample size is 38 participants per group, totaling 76 participants.

5. Revised Manuscript Section:

We will comprehensively revise the "Sample Size" section in the manuscript to explicitly include all parameters, rationale, literature support [1, 2, 4, 7, 8, 9, 10, 11, 12], and the detailed step-by-step calculation shown above.

We appreciate the reviewer's critical input, which has significantly strengthened the methodological rigor of our study protocol. We are confident this revised and fully transparent calculation addresses all concerns raised.

References:

1. Chow S-C, Shao J, Wang H, Lokhnygina Y. Sample Size Calculations in Clinical Research. 3rd ed. Boca Raton: Chapman and Hall/CRC; 2017.

2. Julious SA. Sample Sizes for Clinical Trials. Boca Raton: Chapman and Hall/CRC; 2004.

3. 3. Abbasi Pirouz M, Zojaji A, Shakeri MT, Mirzaei K. Effect of Squill on the sexual function among women of reproductive age. The Iranian Journal of Obstetrics, Gynecology and Infertility. 2018;21(10):57-65.

4. Cohen J. Statistical Power Analysis for the Behavioral Sciences. 2nd ed. Hillsdale, NJ: Lawrence Erlbaum Associates; 1988.

5. Krychman M, Graham S, Bernick B, Mirkin S, Kingsberg SA. The Women's EMPOWER Survey: Women's Knowledge and Awareness of Treatment Options for Vulvar and Vaginal Atrophy Remains Inadequate. J Sex Med. 2017 Mar;14(3):425-433.

6. Portman DJ, Gass ML; Vulvovaginal Atrophy Terminology Consensus Conference Panel. Genitourinary syndrome of menopause: new terminology for vulvovaginal atrophy from the International Society for the Study of Women's Sexual Health and the North American Menopause Society. Menopause. 2014 Oct;21(10):1063-8.

7. Lorenz, T., Rullo, J., & Faubion, S. (2016). Antidepressant-Induced Female Sexual Dysfunction. Mayo Clinic proceedings, 91(9), 1280–1286.

8. Thomas HN, Thurston RC. A biopsychosocial approach to women's sexual function and dysfunction at midlife: A narrative review. Maturitas. 2016;87:49-60. doi:10.1016/j.maturitas.2016.02.009.

9. Isidori AM, Pozza C, Esposito K, Giugliano D, Morano S, Vignozzi L, Corona G, Lenzi A, Jannini EA. Development and validation of a 6-item version of the female sexual function index (FSFI) as a diagnostic tool for female sexual dysfunction. J Sex Med. 2010 Mar;7(3):1139-46.

10. Yule MA, Brotto LA, Gorzalka BB. Biological markers of asexuality: Handedness, birth order, and finger length ratios in self-identified asexual men and women. Archives of Sexual Behavior. 2014 Feb;43(2):299-310.

11. Mitchell CM, Reed SD, Diem S, Larson JC, Newton KM, Ensrud KE, LaCroix AZ, Caan B, Guthrie KA. Efficacy of Vaginal Estradiol or Vaginal Moisturizer vs Placebo for Treating Postmenopausal Vulvovaginal Symptoms: A Randomized Clinical Trial. JAMA Intern Med. 2018 May 1;178(5):681-690

12. Gandhi J, Chen A, Dagur G, Suh Y, Smith N, Cali B, Khan SA. Genitourinary syndrome of menopause: an overview of clinical manifestations, pathophysiology, etiology, evaluation, and management. Am J Obstet Gynecol. 2016 Dec;215(6):704-711

Reviewer Comment 2:

The manuscript does not specify who will generate the randomization sequence, how allocation concealment will be maintained, or what procedures will prevent selection bias.

Author Response:

We thank the reviewer for this important observation. In response, we have revised the Methods section of the manuscript to clearly describe the randomization and allocation procedures. Specifically:

Randomization will be performed using a permuted block design with a block size of 4, generated by computer-based random allocation software (e.g., Random Allocation Software). This method ensures balanced group sizes (1:1 allocation) throughout the recruitment period. An independent statistician will generate the allocation sequence. Group assignment (Myrtus communis gel or placebo) will be concealed using sequentially numbered, opaque, sealed envelopes (SNOSE) until after enrollment. Participants, care providers, outcome assessors, and data analysts will be blinded to group assignment. Additionally, both participants and outcome assessors will be blinded to group assignments to minimize selection and performance bias.

These procedures have been incorporated into the revised manuscript (see Methods section, Sampling and randomization subsection). We believe these modifications adequately address the concerns and help to ensure methodological rigor.

Reviewer Comment 3:

The FSFI cutoff score of <28 should be referenced to validation studies in the Iranian population.

Author Response:

We appreciate the reviewer’s valuable comment. The FSFI cutoff score of <28 has been validated in studies conducted on the Iranian population. Specifically, this cutoff point is supported by references 28 and 29 in our manuscript, which report on the psychometric evaluation of the Persian version of the FSFI and establish a score below 28 as indicative of sexual dysfunction in Iranian women.

Reviewer Comment 3:

The requirement for "at least basic literacy skills" needs operational definition

Author Response:

By “at least basic literacy skills,” we refer to the participants’ ability to read and understand the content of the informed consent form and study questionnaires (e.g., FSFI) without assistance. During the enrollment process, the study coordinator will confirm this by asking each potential participant to read a short paragraph from the consent form aloud and respond to a simple comprehension question.

Reviewer Comment 3:

The criterion of "sexual activity at least once a week" may be too restrictive and could introduce selection bias.

Author Response:

We appreciate the reviewer’s insightful observation. The criterion of “sexual activity at least once a week” was selected considering the one-month duration of the intervention. This criterion was intended to exclude individuals with insufficient sexual activity frequency, in order to enhance the ability to observe the intervention’s effects within the limited study period. We believe this approach helps improve the accuracy of outcome assessment and ensures that participants have adequate exposure to conditions necessary for evaluating sexual function.

Reviewer Comment 3:

The exclusion of women with "vaginal atrophy" requires clear diagnostic criteria

Author Response:

We thank the reviewer for this valuable comment. In response, we have revised the inclusion criteria to include clear diagnostic indicators for excluding women with vaginal atrophy.

Reviewer Comment 4:

Discuss the following: (1) informed consent procedures and content, (2) procedures for obtaining consent from participants with limited literacy, (3) confidentiality protections given the sensitive nature of sexual health research, and (4) procedures for managing participants who develop concerning symptoms during the study.

Author Response:

Thank you for your valuable comment. The requested information regarding (1) informed consent procedures and content, (2) procedures for obtaining consent from participants with limited literacy, (3) confidentiality protections considering the sensitive nature of sexual health research, and (4) procedures for managing participants who develop concerning symptoms during the study has been addressed and incorporated under the "Ethical Considerations" section of the revised manuscript.

Reviewer Comment 5:

5- Lines 267-275: Statistical analysis section needs expansion to include missing data handling, interim analyses, and multiple comparison procedures.

Author Response:

We thank the reviewer for this valuable comment. In response:

Primary Analysis: Intention-to-treat (ITT) analysis will be performed, including all randomized participants in their originally assigned groups, regardless of adherence or dropout.

Missing Data Handling (ITT): For the primary outcome (FSFI change), missing post-intervention data will be addressed using multiple imputation by chained equations (MICE), creating 20 imputed datasets. The imputation model will include baseline FSFI score, age, group assignment, and relevant covariates showing association with missingness. Sensitivity analyses will compare results to complete-case analysis.

Supportive Analysis: A per-protocol (PP) analysis will also be conducted, including only participants meeting predefined adherence criteria (≥80% adherence) and completing the 1-month assessment. Results from ITT and PP analyses will be compared to assess robustness.

Reviewer Comment 6:

6- Lines 185-205: Inclusion criteria should be justified or reconsidered, particularly the literacy requirement and weekly sexual activity criterion

Author Response:

We appreciate your comment. The inclusion criterion of basic literacy was necessary to ensure that participants could understand the informed consent form and independently complete self-report questionnaires such as the FSFI. The requirement for weekly sexual activity was based on the structure of the FSFI, which evaluates function over the past 4 weeks; participants who are not sexually active would not be able to provide valid responses, which could compromise data quality. These justifications have now been added to the Inclusion Criteria section.

---

## [Decision Letter · Decision Letter 1]

5 Nov 2025

Dear Dr. babazadeh,

Thank you for submitting your manuscript to PLOS ONE. After careful consideration, we feel that it has merit but does not fully meet PLOS ONE’s publication criteria as it currently stands. Therefore, we invite you to submit a revised version of the manuscript that addresses the points raised during the review process.

1- sample size:

The sample size calculation presented in the manuscript contains several concerning issues that require immediate attention. The authors reference a study by Abbasi Pirouz et al. reporting mean differences of 24.5 ± 4.5 in the intervention group and 3.4 ± 5.8 in the control group, leading to a calculated sample size of "two participants using the formula for the mean of two independent populations." This calculation appears to be fundamentally flawed.

First, the formula presented (n = 2(z₁₋α + z₁₋β)²/f²) is incomplete and lacks proper definition of the effect size parameter 'f'. The authors then state they calculated an effect size of 0.7 and arrived at 32 participants per group, but the mathematical steps are not shown or justified. The jump from 2 participants to 32 participants per group is not explained, and the 20% dropout rate adjustment to reach 38 participants per group lacks proper justification.

The authors should provide a complete, step-by-step sample size calculation including: (1) clearly defined primary outcome and clinically meaningful difference, (2) expected standard deviation based on pilot data or literature, (3) specified Type I and Type II error rates, (4) detailed calculation showing all mathematical steps, and (5) justification for the dropout rate assumption.

2- The manuscript does not specify who will generate the randomization sequence, how allocation concealment will be maintained, or what procedures will prevent selection bias.

3- The FSFI cutoff score of <28 should be referenced to validation studies in the Iranian population

•The requirement for "at least basic literacy skills" needs operational definition

•The criterion of "sexual activity at least once a week" may be too restrictive and could introduce selection bias

•The exclusion of women with "vaginal atrophy" requires clear diagnostic criteria

4- Discuss the following: (1) informed consent procedures and content, (2) procedures for obtaining consent from participants with limited literacy, (3) confidentiality protections given the sensitive nature of sexual health research, and (4) procedures for managing participants who develop concerning symptoms during the study.

5- Lines 267-275: Statistical analysis section needs expansion to include missing data handling, interim analyses, and multiple comparison procedures.

6- Lines 185-205: Inclusion criteria should be justified or reconsidered, particularly the literacy requirement and weekly sexual activity criterion.

We look forward to receiving your revised manuscript.

Kind regards,

Akingbolabo Daniel Ogunlakin, Phd

Academic Editor

PLOS ONE

Journal Requirements:

Additional Editor Comments:

1- sample size:

The sample size calculation presented in the manuscript contains several concerning issues that require immediate attention. The authors reference a study by Abbasi Pirouz et al. reporting mean differences of 24.5 ± 4.5 in the intervention group and 3.4 ± 5.8 in the control group, leading to a calculated sample size of "two participants using the formula for the mean of two independent populations." This calculation appears to be fundamentally flawed.

First, the formula presented (n = 2(z₁₋α + z₁₋β)²/f²) is incomplete and lacks proper definition of the effect size parameter 'f'. The authors then state they calculated an effect size of 0.7 and arrived at 32 participants per group, but the mathematical steps are not shown or justified. The jump from 2 participants to 32 participants per group is not explained, and the 20% dropout rate adjustment to reach 38 participants per group lacks proper justification.

The authors should provide a complete, step-by-step sample size calculation including: (1) clearly defined primary outcome and clinically meaningful difference, (2) expected standard deviation based on pilot data or literature, (3) specified Type I and Type II error rates, (4) detailed calculation showing all mathematical steps, and (5) justification for the dropout rate assumption.

2- The manuscript does not specify who will generate the randomization sequence, how allocation concealment will be maintained, or what procedures will prevent selection bias.

3- The FSFI cutoff score of <28 should be referenced to validation studies in the Iranian population

•The requirement for "at least basic literacy skills" needs operational definition

•The criterion of "sexual activity at least once a week" may be too restrictive and could introduce selection bias

•The exclusion of women with "vaginal atrophy" requires clear diagnostic criteria

4- Discuss the following: (1) informed consent procedures and content, (2) procedures for obtaining consent from participants with limited literacy, (3) confidentiality protections given the sensitive nature of sexual health research, and (4) procedures for managing participants who develop concerning symptoms during the study.

5- Lines 267-275: Statistical analysis section needs expansion to include missing data handling, interim analyses, and multiple comparison procedures.

6- Lines 185-205: Inclusion criteria should be justified or reconsidered, particularly the literacy requirement and weekly sexual activity criterion.

Reviewers' comments:

Reviewer's Responses to Questions

**Comments to the Author**

1. Does the manuscript provide a valid rationale for the proposed study, with clearly identified and justified research questions?

Reviewer #1: Yes

Reviewer #2: Yes

Reviewer #4: Yes

2. Is the protocol technically sound and planned in a manner that will lead to a meaningful outcome and allow testing the stated hypotheses?

Reviewer #1: No

Reviewer #2: Yes

Reviewer #4: Yes

3. Is the methodology feasible and described in sufficient detail to allow the work to be replicable?

Reviewer #1: No

Reviewer #2: Yes

Reviewer #4: Yes

4. Have the authors described where all data underlying the findings will be made available when the study is complete?

Reviewer #1: Yes

Reviewer #2: Yes

Reviewer #4: Yes

5. Is the manuscript presented in an intelligible fashion and written in standard English?

Reviewer #1: Yes

Reviewer #2: Yes

Reviewer #4: Yes

You may also provide optional suggestions and comments to authors that they might find helpful in planning their study.

Reviewer #1: They have provided the information for the sample size calculation which was one of my concerns noted to the editor. The effect size of 0.7 was a conservative estimate from their original effect size which I believe was 4.07 based on their data.

According to the investigators in the track changes they note that “ In accordance with the reviewers' :Commented [RB(6] comments, Figure 1 has been removed from the main text and is now provided exclusively as S1 Figure in the ".Supporting Information file”. I still do not see this Figure S1 in this file.

The English translation of the protocol is now in the text I believe.

Reviewer #2: The authors have satisfactorily addressed all my concerns, and the manuscript is now methodologically sound and clearly written. I recommend acceptance in its current form.

Reviewer #4: This manuscript is well written and very insightful it will contribute to knowledge enormously. A better manuscript would be produced if some of the following errors are corrected majorly with sentence structures:

1. Study Setting. Line 147 should be written in past tense

2. Recruitment. Lines 150-155 should be written in past tense. similar observations were made in Sampling, Lines 212-226, Interventions and control, Lines 229-250, Blinding, Lines 269-275, Adherence Monitoring and Missing Data Management, Lines 278-283, Ethical Consideration, Lines 286-301 and Statistical Methods Lines 305-323

**Do you want your identity to be public for this peer review?** For information about this choice, including consent withdrawal, please see our Privacy Policy

Reviewer #1: No

Reviewer #2: No

Reviewer #4: No

---

## [Author Response · Author response to Decision Letter 2]

17 Nov 2025

Dear Editor,

Ref:

Re: " The effects of vaginal gel from Myrtus communis on the sexual function of married women during reproductive aging: a study protocol for a randomized controlled trial "

I appreciate very much the comments of reviewers toward the improvement of the paper and I hope that a suitable revision to addresses reviewers' comments is carried out. Responses to the comments/questions of the reviewers along with a description of the changes made on the manuscript are given below. The amended parts have been highlighted in YELLOW in the manuscript.

I would be glad to respond to any further questions and comments that you may have.

Sincerely yours,

Corresponding author

Dr. Raheleh Babazadeh

Comments from Reviewers

Comment 1: Line 147 should be written in past tense.

Response: Line 147 has been changed to past tense.

Comment 2: Recruitment. Lines 150-155 should be written in past tense. Similar observations were made in Sampling, Lines 212-226, Interventions and control, Lines 229-250, Blinding, Lines 269-275, Adherence Monitoring and Missing Data Management, Lines 278-283, Ethical Consideration, Lines 286-301 and Statistical Methods Lines 305-323

Response: As requested, lines 150–155 have been changed to past tense. The tense of the sentences in Sampling (Lines 212–226), Interventions and Control (Lines 229–250), Blinding (Lines 269–275), Adherence Monitoring and Missing Data Management (Lines 278–283), Ethical Consideration (Lines 286–301), and Statistical Methods (Lines 305–323) has been changed to the past tense.

---

## [Decision Letter · Decision Letter 2]

26 Nov 2025

The effects of vaginal gel from Myrtus communis on the sexual function of married women during reproductive aging: a study protocol for a randomized controlled trial

PONE-D-25-17029R2

Dear Dr. raheleh babazadeh,

We’re pleased to inform you that your manuscript has been judged scientifically suitable for publication and will be formally accepted for publication once it meets all outstanding technical requirements.

Kind regards,

Akingbolabo Daniel Ogunlakin, Phd

Academic Editor

PLOS ONE

Additional Editor Comments (optional):

Reviewers' comments:

Reviewer's Responses to Questions

**Comments to the Author**

1. Does the manuscript provide a valid rationale for the proposed study, with clearly identified and justified research questions?

Reviewer #1: Yes

2. Is the protocol technically sound and planned in a manner that will lead to a meaningful outcome and allow testing the stated hypotheses?

Reviewer #1: Yes

3. Is the methodology feasible and described in sufficient detail to allow the work to be replicable?

Reviewer #1: Yes

4. Have the authors described where all data underlying the findings will be made available when the study is complete?

Reviewer #1: Yes

5. Is the manuscript presented in an intelligible fashion and written in standard English?

Reviewer #1: Yes

You may also provide optional suggestions and comments to authors that they might find helpful in planning their study.

Reviewer #1: Figure S1 was included as I requested.

XXXXXXXXXXXXXXXXXXXXXXXXXXXXXXXXXXXXXXXXXXXXXXXXXXXXXXXXXXXXXXXXXXXXXXXXXXXXXXXXXXXXXXXXXXXXXXXXXXXXXXXXXXXXXXXXXXXXXXXXXXXXX

**Do you want your identity to be public for this peer review?** For information about this choice, including consent withdrawal, please see our Privacy Policy

Reviewer #1: No

---

## [Editor Report · Acceptance letter]

PONE-D-25-17029R2

PLOS One

Dear Dr. babazadeh,

I'm pleased to inform you that your manuscript has been deemed suitable for publication in PLOS One. Congratulations! Your manuscript is now being handed over to our production team.

Kind regards,

on behalf of

Dr. Akingbolabo Daniel Ogunlakin

Academic Editor

PLOS One